# Time-to-Treatment of Oral Cancer and Potentially Malignant Oral Disorders: Findings in Malaysian Public Healthcare

**DOI:** 10.3390/dj10110199

**Published:** 2022-10-24

**Authors:** Sivaraj Raman, Asrul Akmal Shafie, Mannil Thomas Abraham, Shim Chen Kiong, Thaddius Herman Maling, Senthilmani Rajendran, Sok Ching Cheong

**Affiliations:** 1Centre for Health Economics Research, Institute for Health Systems Research, National Institutes of Health, Shah Alam 40170, Malaysia; 2Discipline of Social and Administrative Pharmacy, School of Pharmaceutical Sciences, Universiti Sains Malaysia, Penang 11800, Malaysia; 3Institutional Planning and Strategic Center, Universiti Sains Malaysia, Penang 11800, Malaysia; 4Oral and Maxillofacial Surgery Department, Hospital Tengku Ampuan Rahimah, Ministry of Health, Klang 41200, Malaysia; 5Oral and Maxillofacial Surgery Department, Hospital Umum Sarawak, Ministry of Health, Kuching 93586, Malaysia; 6Samarahan Divisional Dental Office, Sarawak State Health Department, Ministry of Health, Kota Samarahan 94300, Malaysia; 7Digital Health Research Unit, Cancer Research Malaysia, Subang Jaya 47500, Malaysia; 8Department of Oral and Maxillofacial Clinical Sciences, Faculty of Dentistry, University of Malaya, Kuala Lumpur 50603, Malaysia

**Keywords:** delays, time-to-treatment, oral cancer, precancerous conditions, public hospitals, access, risk-taking

## Abstract

This study aims to evaluate the time-to-treatment of oral cancer and potentially malignant oral disorders (PMOD) in a Malaysian public healthcare setting while exploring its contributing factors. It consists of (1) a cross-sectional patient survey to quantify time to seek care and barriers faced, and (2) a retrospective medical record abstraction to determine treatment and management intervals. Time intervals were aggregated and analyzed by their primary contributor—patient, professional, or healthcare system. The average total time-to-treatment of the 104 patients investigated was 167 days (SD = 158). This was predominantly contributed by the patient interval of 120 days (SD = 152). In total, 67.0% of patients delayed their visit to primary healthcare centers because they assumed the lesions were not dangerous or of concern. Additionally, there was a significant difference between patients ‘facing’ and ‘not facing’ difficulties to seek care, at 157 vs. 103 days (*p* = 0.028). System and professional delays were comparably shorter, at 33 days (SD = 20) and 10 days (SD = 15) respectively. Both demonstrated a significant difference between oral cancer and PMOD, at 43 vs. 29 days (*p* < 0.001) and 5 vs. 17 days (*p* < 0.001). The findings reiterate the need to reform current initiatives to better promote early lesion recognition by patients and implement strategies for the elimination of their access barriers.

## 1. Introduction

Oral cancer is one of the most common cancers in the Asian region, totaling 240,736 incidences and 128,799 mortalities in 2020 alone [1]. While these incidences contributed to 65.8% of worldwide cases, the mortality rates were disproportionately higher at 74% [1]. This reflects a greater threat of oral cancer in the region, which is believed to be precipitated by factors such as lower socioeconomic conditions and cultural habits such as betel quid chewing and smoking [2]. In Malaysia, oral cancer remains within the top 20 most common cancers with a five-year prevalence of 2199 cases. Worryingly, the incidence of oral cancer is projected to double by the year 2040 following the local age-specific incidence and population growth [3].

Oral cancer can arise in either seemingly healthy mucosa or be preceded by abnormal lesions categorized as potentially malignant oral disorders (PMOD). They consist of a myriad of diagnoses such as oral leukoplakia, erythroplakia, submucous fibrosis, and lichen planus [4]. These abnormalities vary largely by their clinical presentation, symptoms, and the corresponding risk of malignant transformation. Although both PMOD and oral cancer are generally conspicuous, patients are still predominately diagnosed in the later stages of cancer [5]. Consequently, their survival rates drop drastically as cancer spreads to regional lymph nodes and metastasizes [4]. Such a trend was postulated to be the main reason for the mortality rates remaining plateaued at 50% in the past three decades, despite continued advancements in treatment modalities [4]. 

To abate such risks, the Malaysian National Oral Health Plan mapped out an initial minimum target of 30% of oral cancer cases to be detected at stage I [6]. However, after almost a decade of implementation of various initiatives, the goal is yet to be realized. The current national cancer registry data shows a grim reality of only 15.6% of all oral cancer cases being diagnosed at stage I [7]. The lower percentages are suggested to be attributed to diagnostic delays stemming from poor patient awareness and knowledge, especially in lower sociodemographic populations. Inequities and accessibility barriers to dental services are also recognized as possible contributing factors [8]. In addition to patient factors, a range of issues from lack of experience and vigilance by the primary healthcare personnel, to congestion in secondary or tertiary centers could have compounded such late diagnoses [9].

As the incidence of oral cancer and PMOD are anticipated to surge continuously, this study aims to quantify the time from the appearance of symptoms to the initiation of treatment while exploring the contributing factors. This evidence is vital to realign the current focus of national initiatives to better safeguard the public. Malaysia undoubtedly forms an interesting case study to identify possible gaps in a highly subsidized public healthcare system [10].

## 2. Materials and Methods

A cross-sectional study was conducted among patients with oral cancer and PMOD attending the oral maxillofacial specialist clinics in Hospital Tengku Ampuan Rahimah, Klang and Hospital Umum Sarawak, Kuching, Malaysia. Both the public tertiary hospitals were selected as they are the main referral centers in west and east Malaysia, catering to a large and diverse population. The study consisted of (1) a retrospective patient survey to quantify the time taken to seek care and the barriers faced; and (2) a medical record abstraction to determine professional and system delays involved in their care management. 

### 2.1. Sampling

The sample size was calculated to detect an expected mean difference of 14 days with a standard deviation of 21 days in a healthcare system delay between oral cancer and PMOD, as part of the analysis explored the impact by diagnosis groups [11,12]. The sample size calculation formula applied appears below:$$n\;=\;\left[{\left(Z_{\alpha /2}+Z_{\beta }\right)}^{2}\times 2\;\times \sigma^{2}\right]÷d^{2}$$

Z_α/2_ = critical value of the normal distribution at α/2, where α was set at 0.05; Z_β_ = critical value of the normal distribution at β, where β was set at 0.2; σ^2^ = population variance; d = differences in mean to be detected.

Based on a confidence level of 95% and precision of 80%, the calculated sample required was 36 per group. A convenience sampling of adult patients (over 18 years of age) attending their routine outpatient review was adopted. The sampling was stratified by diagnosis to ensure equal representation of cases with PMOD and various stages of oral cancer. 

To capture the details of the patient journey from diagnosis to treatment, only those who had been diagnosed histologically with either dysplasia or oral squamous cell carcinoma and started receiving treatment were recruited. This consisted of the initiation of treatments, such as surgical or oncologic interventions in oral cancer, or excisional surgery and oral/topical medications in PMOD. Oral cancer was defined according to the International Classifications of Diseases 10th revision, ranging from C00 to C06 [13]. Oral cancer staging followed the American Joint Committee on Cancer (AJCC) TNM system, based on the extent of the primary tumor (T), regional lymph node involvements (N), and the presence of metastasis (M). Conversely, PMOD was defined according to the 2007 consensus by the WHO Collaborating Centre for Oral Cancer and Precancer [14].

### 2.2. Defining Time-to-Treatment

The definitions and characterization of delays in cancer treatment are variable and complex in nature. To standardize methodological approaches and study outcomes, the Aarhus Statement by an international working group was applied to guide the reporting of time–point measurements [15]. The time intervals (in days) between defined events were used as proxies for delays and adapted according to the clinical framework in Malaysian public healthcare. The time-to-treatment intervals were further divided into three categories—patient, professional, and healthcare system—according to the primary contributor of the wait time, to allow for meaningful analysis (Table 1) [16].

### 2.3. Patient Survey

Participants were briefed about the study and informed written consent was obtained. They were then required to complete a three-part interviewer-guided survey consisting of (1) sociodemographic data, (2) the duration of time to seek oral examination and referral to a specialist, and (3) an exploration of contributing factors to their delays. The primary reason for hesitancy to seek treatment was inquired through an open-ended question to ensure the comprehensiveness of the findings. Other factors, such as barriers to primary care for oral examinations, presence of health screening fears, oral cancer awareness, frequency of routine dental check-ups, and confidence in oral cancer screening by dentists, were also explored in the third part of the survey. 

### 2.4. Medical Record Abstraction

The patient survey was followed up by tracing their medical records and abstracting information using standardized proforma. This included clinical information, such as diagnosis, staging, primary lesion sites, and treatment modalities. The proforma was used to report the date of events and to calculate their respective time interval. Recording of available dates was conducted according to the hierarchy of priority defined by the European Network of Cancer Registries [15]. Additionally, the patient-reported duration of seeking care in the earlier survey was triangulated with documented history in medical records whenever possible to minimize recall biases [15].

### 2.5. Statistical Analysis

The time intervals were reported in both mean (SD) and median (interquartile ranges) to ensure the applicability of findings for decision-making and future analysis. The values were aggregated according to contributors and investigated for differences between diagnosis of PMOD and oral cancer using the Kruskal–Wallis test, with significance set at *p* < 0.05. The Kruskal–Wallis test was also used to explore the impact of socio-demographic factors, such as income, patient barriers and fears, health-seeking behaviors, and other clinical factors with relevant time intervals. The patient’s household income was categorized as either below or above a threshold of MYR 4360 (USD 2722). This was the cut-off for the lower 40th percentile of the national median monthly income, which is the commonly applied delineation for national policies [17]. The conversion rate was based on the purchase power parity in the year 2019 (1 USD = 1.602 MYR). 

## 3. Results

A total of 104 patients with oral cancer and PMOD were surveyed, and their medical records were successfully abstracted. Those diagnosed with oral cancer consisted of stage I (*n* = 2, 3.8%); stage II (*n* = 8, 15.4%); stage III (*n* = 13, 25.0%); and stage IV (*n* = 29, 55.8%), mirroring the larger late-stage diagnosis in national reports. A diagnosis of lichen planus formed the majority of patients with PMOD at 58.0%, followed by leukoplakia at 16.0%. The sociodemographic and clinical characteristics of the subjects are reported in Table 2. There were no significant differences in the characteristics between patients with oral cancer and PMOD, except in education. A Fisher’s exact test of independence showed there was a significantly larger proportion of patients with lower education levels in the oral cancer group relative to PMOD. While there was no difference in the income category, a subgroup analysis showed there was a significant difference in the mean monthly household income between PMOD and oral cancer, at MYR 2520 and MYR 1988 respectively, χ^2^ (1, *n* = 104) = 6.3, *p* = 0.012.

### 3.1. Time-to-Treatment

The overall mean time-to-treatment was 167 days (SD = 158), with a median of 103 days (IQR = 67–211). This ranged from 16 to 798 days for PMOD and 38 to 761 days for oral cancer. The average patient interval of 120 days (SD = 149) formed 73.2% of the total time-to-treatment, followed by the system interval of 33 days (SD = 20), which contributed to 21.5% of the duration, and the professional interval of 10 days (SD = 14) at 5.3%. Table 3 shows the comparison of intervals by diagnosis groups to avoid generalization. The Kruskal–Wallis test shows that the differences between PMOD and oral cancer were only seen in professional and healthcare system intervals.

### 3.2. Factors Influencing the Patient Interval

Further analysis showed the patient interval was not significantly associated with diagnosis groups, tumor sites, or the reported sociodemographic factors (Appendix A). In total, 67.0% of patients delayed their visit to primary healthcare centers because they assumed the lesions were not dangerous or of concern: 26.8% claimed lesions were painless, 23.7% assumed normalcy, and 16.5% believed lesions will heal naturally. The rest consisted of 19.6% attributing delays to other factors, and only 13.4% were identified opportunistically. There was a significantly longer interval from the onset of symptoms to the first healthcare visit (T_1_) in the group assuming lesions were non-malignant (141 days) relative to those indicating other factors (90 days), and were opportunistically identified by clinicians (19 days) with *χ*^2^ (2) = 20.1, *p* < 0.001.

The findings on possible barriers and fears faced by patients in accessing initial care are shown in Figure 1. There was a significant difference in duration of T_1_ between patients reporting no difficulties and those facing any difficulties to access primary care for oral examinations, at 103 vs. 157 days, *χ*^2^ (1) = 0.7, *p* = 0.028. On the other hand, there was no difference in T_1_ between patients reporting no fears and presence of any fears, at 119 and 123 days, *χ*^2^ (1) = 4.8, *p* = 0.402.

The last area evaluated was the awareness of concerning symptoms and health-seeking behaviors of patients. Slightly over half of the patients surveyed (59.6%) had heard about oral cancer before their diagnosis, while only 16.5% of them had prior annual dental check-ups. Interestingly, 46.5% of patients did not agree that dentists are trained for oral cancer screening and only 16.5% had prior annual dental check-ups. However, all these factors were not significantly associated with T_1_ (Appendix A).

### 3.3. Factors Influencing Professional and System Intervals

A total of 56.1% of patients were first seen by general practitioners for their lesions, while the rest were first seen by dentists. There was no difference in the interval for specialist referral (T_2_) between both practitioners, at 4 vs. 5 days respectively, *χ*^2^ (1) = 0.1, *p* = 0.730. The system interval showed no significant differences between lesion sites (buccal mucosa, tongue, and others), or between oral cancer stages (stage I/II and stage III/IV). Similarly, there was also no difference in the interval between oral cancer treatments, including radiotherapy, chemotherapy, or reconstructive surgeries (*p* > 0.05).

## 4. Discussion

This study reaffirms the presence of delays in diagnosis for PMOD and oral cancer, despite the numerous national initiatives implemented in Malaysia. The average time-to-treatment was almost half a year, with some cases going beyond two years. Such prolonged intervals can impact patients’ survival and prognosis, as it allows tumors to enlarge and metastasize [12,18]. While no evidence is currently available to demarcate a specific delay duration for increased mortality risks, Lopez et al. (2020) found an association between the time interval and mortality rates among patients with oral cancer. They reported patients with a longer treatment interval (128–420 days) had a higher mortality risk, relative to those in the middle range (56–128 days) [19]. 

The average patient interval of around 120 days in Malaysia can be categorized as excessively lengthy. This is because most guidelines advise patients to seek professional care if abnormal lesions or symptoms persist beyond two to three weeks [20]. Such a trend was, however, comparable to studies around the world, as shown in Table 4 [12,20,21,22,23,24]. The only reassuring prospect is that our current findings show a reduction in the patient interval in Malaysia when compared with values reported by Khoo et al. from the past two decades (Table 4) [21]. Even though the decrement is commendable, the current pace remains insufficient to generate meaningful changes in the national oral cancer burden, as reflected by the high prevalence of late-stage diagnoses [7].

Patient delay, often described as primary delay, can be interpreted as a form of health risk-taking behavior. It is believed to be influenced and motivated by multiple factors, ranging from awareness to socioeconomic and clinical characteristics [25]. Oddly, none of the sociodemographic variables, tumor sites, or severity of diagnosis were associated with our patients’ health-seeking behavior. This was, nonetheless, in agreement with several recent reports in both western and Asian countries [12,23]. Patient intervals were postulated to be largely related to psychosocial and cognitive factors [12]. While local investigations are still warranted to identify possibilities of confounders, the current evidence highlights that these components need not be the primary focus and determinant in structuring national oral cancer campaigns.

Difficulties in accessing care due to factors such as financial and logistical challenges are some of the more critical areas that can be explored for improvements as they influence our patient delay. The public healthcare system in Malaysia is highly subsidized with patient fees only compensating 2 to 3% of the Ministry of Health’s total expenditure [26]. Despite this, accessibility may still be a challenge for patients with lower incomes, especially those requiring longer journeys to obtain professional care. Furthermore, dental clinics in Malaysia are highly concentrated in major cities and among relatively richer populations [27]. For example, the distribution of dental clinics in rural areas in east Malaysia was one of the lowest, despite having a high oral disease burden [27]. Thus, accessibility to care should be considered in devising oral cancer screening and preventive programs. Implementation of more frequent community screening visits and incorporation of e-health strategies may narrow these health inequities without a large economic burden to the ministry.

One of the most important findings from our study is that the diagnosis group (PMOD or oral cancer) did not alter the patient’s hesitancies in seeking help. This was surprising, as lesions and symptoms in oral cancer are generally more debilitating relative to PMOD, thus expected to compel faster help-seeking behavior. This trend was also seen in countries including Taiwan and Japan [28,29]. Such behavior can be explained by a self-regulatory model, which proposes patients often misinterpret oral cancer symptoms as induced by factors such as food or injuries [30]. Consequently, patients selectively adopt inappropriate behavioral responses and assume lesions will resolve over time. The misattribution of symptoms frequently leads to hesitancy in seeking professional advice, regardless of the severity. This was further evidenced in our sample as the patients corroborated their initial nonchalant attitude to being driven by assumptions of normalcy and lack of pain.

Even though dentists are trained to detect PMOD during dental care exams, the lack of routine visits may further delay the diagnosis. Interestingly, only about half of the patients surveyed opined that dentists are trained to conduct oral examinations and investigate signs of oral cancer. This belief may have partly contributed to a slightly larger proportion of our respondents seeking general practitioners as their first reference point [31]. Such a trend was also reported in developed nations. Patients were described to favor general practitioners over a dentist for oral examinations with the presence of worrying signs and symptoms [31]. Thus, general practitioners should also be engaged as part of the National Oral Health Plan and undergo frequent training in the diagnosis and care pathways of oral cancer [6]. This is suggested because a recent systematic review identified symptom misinterpretation by general practitioners as a common reason contributing to professional delay at the primary care level [32]. Nevertheless, there was no significant difference in the referral time to specialists by both types of practitioners in our sample.

For both professional and system intervals, the only significant difference between oral cancer and PMOD was the disease severity. In the professional interval, the duration of time taken for the referral to oral specialists and the biopsy was shorter for patients eventually diagnosed with oral cancer compared with PMOD. This primarily reflects the urgency of care needed for patients with larger lesions and more debilitative symptoms. The finding is consistent with oral cancers being diagnosed earlier relative to smaller lesions [25,28]. Due to the referral system in Malaysian public healthcare, patients with severe diseases can be referred directly to tertiary centers. On the other hand, lesions that are not easily identified or assumed to lack urgency may take the longer path through the referral chain. The longer duration for PMOD may also be explained by the exploration of treatment options such as antibiotics, or investigations before referrals to tertiary centers [11].

System intervals, on the contrary, were largely contributed to by both the decision plan and the treatment regimens. In oral cancer, the treatment plan involves a multidisciplinary team consisting of the oral pathologist, maxillofacial surgeons, plastic surgeons, and oncologists [33]. The assembly of the team of specialists undeniably takes a longer period relative to PMOD, where decisions are often made solely by the examining specialists. Equally, the delays involved in the allocation of resources, such as the availability of personnel and operating theaters, radiotherapy sessions, preparation, and administration of chemotherapies in an already congested public healthcare system are currently unavoidable. These delays were shown to occur regardless of the treatment modality. Improvement in this area involves a larger commitment from the ministry and possibly public-private partnerships to increase the capacity for medical staff and facilities to treat oral cancer.

While our findings on the delay intervals and factors are invaluable in the region, their interpretations should be cautiously constructed by considering several limitations. The main limitation arises from the exploratory approach and categorization of contributing factors associated with treatment delays. The primary factor, the effect of socio-demographic confounders and other enabling influences, was not captured in this survey. Thus, a further study with in-depth patient interviews is warranted to ascertain if such associations exist. Secondly, the sampling in public healthcare facilities may skew responses toward compliant patients and those from middle to lower socioeconomic groups. Nevertheless, as the majority of the population receives care in public healthcare facilities compared to private, these values still provide insight into the implementation of national oral health policies.

## 5. Conclusions

The prolonged time-to-treatment of oral cancer and PMOD in Malaysia is still primarily contributed to by the time taken by patients to seek care. There is a dire need for public awareness campaigns to educate the public to recognize abnormal lesions as potentially malignant and encourage early self-referral. Furthermore, accessibility to primary care centers for oral examinations and early screenings should also be investigated by policymakers to remove barriers and improve participation. As both population and oral cancer incidences are projected to increase over time, healthcare professionals may need to place more emphasis on accelerating the referral process and optimizing resources for treatment. Increasing the capacity of skilled healthcare personnel and upscaling facilities are warranted to avoid future congestion and potential delays in public healthcare.

## Figures and Tables

**Figure 1 dentistry-10-00199-f001:**
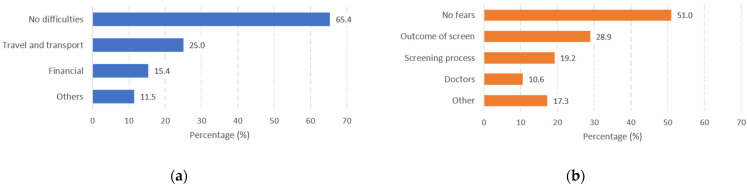
(**a**) Percentage of respondents reporting difficulties in accessing initial care; (**b**) percentage of respondents reporting fears towards the oral examination. Patients were allowed to choose more than one related factor.

**Table 1 dentistry-10-00199-t001:** Breakdown of intervals in time-to-treatment.

Interval ^1^	Process	Duration (Days)
Patient	T_1_	From the time the patient first became aware of symptoms to the first visit to primary care
Professional	T_2_	From the first visit to primary care to the first referral to a specialist
Patient	T_3_	From the appointment date to actual attendance at the specialist clinic
Professional	T_4_	From the first exam by a specialist to the period a biopsy was first performed
System	T_5_	From the first biopsy taken to the reporting of biopsy results
System	T_6_	From the reporting of the biopsy results to the assignment of a treatment plan by a multidisciplinary oral cancer team
System	T_7_	From the assignment of a treatment plan to the initiation of definitive treatment (first day of radiotherapy, chemotherapy, surgery, or oral/topical treatment)

^1^ Primary contributor to the interval was based on Olesen, Hansen, and Vedsted [16].

**Table 2 dentistry-10-00199-t002:** Sociodemographic and clinical characteristics of patients.

Characteristic		PMOD (*n* = 52) Freq (%)	Cancer (*n* = 52) Freq (%)	*p*-Value ^1^
Age	<60	25 (48.1)	26 (50.0)	0.844
	>60	27 (51.9)	26 (50.0)	
Gender	Male	18 (34.6)	24 (46.2)	0.230
	Female	34 (65.4)	28 (53.8)	
Race	Malay	14 (26.9)	7 (13.5)	0.110
	Chinese	6 (11.5)	14 (26.9)	
	Indian	27 (51.9)	24 (46.1)	
	Indigenous	5 (9.6)	7 (13.5)	
Location	Urban	26 (50.0)	23 (44.2)	0.556
	Rural	26 (50.0)	29 (55.8)	
Education	None/Primary	18 (34.6)	30 (57.7)	0.018
	Secondary/Tertiary	34 (65.4)	22 (42.3)	
Occupation	Not employed	21 (40.4)	23 (44.2)	0.691
	Employed/Retired	31 (59.6)	29 (55.8)	
Household income	≤MYR 4360	47 (90.4)	46 (88.5)	0.750
	>MYR 4360	5 (9.6)	6 (11.5)	
Anatomic site	Buccal mucosa	32 (61.5)	25 (48.1)	0.408
	Tongue	13 (25.0)	14 (26.9)	
	Others ^2^	7 (13.5)	13 (25.0)	

^1^ Difference between patients with PMOD and oral cancer based on sociodemographic characteristics using Fisher’s exact test, with significance set to *p* < 0.05. ^2^ Consists of the alveolar, gingiva, lip, the floor of mouth, palate, mandible, and other sites.

**Table 3 dentistry-10-00199-t003:** Breakdown of time-to-treatment (days) by diagnosis.

Interval	Duration (Days)				*p*-Value ^1^
	PMOD (*n* = 52)		Cancer (*n* = 52)		
	Mean (SD)	Median (IQR)	Mean (SD)	Median (IQR)	
Patient	111 (143)	64 (15–120)	128 (155)	61 (21–183)	0.7616
T_1_: Symptom to the first primary care visit	106 (140)	60 (15–105)	130 (156)	60 (23–180)	0.5075
T_3_: Specialist appointment to actual clinic attendance	5 (10)	1 (0–4)	1 (3)	0 (0–2)	0.0564
Professional	15 (17)	8 (2–20)	4 (8)	1 (0–6)	0.0001
T_2_: First primary visit to specialist referral	7 (12)	1 (0–8)	2 (7)	0 (0–1)	0.0046
T_4_: Specialist visit to biopsy	9 (14)	5 (0–11)	2 (4)	0 (0–1)	0.0006
System	25 (18)	21 (11–37)	42 (19)	41 (29–50)	0.0001
T_5_: First biopsy to biopsy results	11 (6)	10 (6–16)	9 (5)	8 (6–12)	0.1211
T_6_: Biopsy results to treatment plan	11 (10)	9 (1–16)	19 (12)	15 (9–29)	0.0009
T_7_: Treatment plan to treatment initiation	6 (12)	0 (0–13)	14 (11)	11 (6–20)	0.0001
TOTAL delay	157 (154)	109 (68–171)	176 (163)	95 (67–230)	0.9626

^1^ Difference between patients with PMOD and oral cancer based on intervals using the Kruskal–Wallis H test, with significance set to *p* < 0.05.

**Table 4 dentistry-10-00199-t004:** Summary of time-to-treatment in literature and the recommended intervals.

Author	Year	Population	Interval Period (in Days)		
			Patient	Professional	Healthcare
Study findings	2021	Malaysia	120	10	33
Khoo et al. [21]	1996	Malaysia	202	72 ^1^	-
Varela-Centelles et al. [22]	2017	Europe, USA, India, Australia, Japan, Argentina, Iran	80	16	59
Saka-Herrán et al. [12]	2021	USA, Germany, China, Europe, Iran, India, Australia, Japan, Argentina, Canada, Denmark	48–168	14–90 ^2^	29–57 ^2^
Stefanuto et al. [23]	2014	Germany, USA, Canada, UK, Thailand, Japan, the Netherlands	105–162	98–147	
Available recommendations					
NICE Guideline (NG12) [20]	2021	United Kingdom	0–21 ^3^	14 ^4^	-
Brazil Federal Law [24]	2012	Brazil	-	30 ^5^	60 ^5^

^1^ Mean duration for definitive diagnosis by clinicians. ^2^ Based on median interval. ^3^ Immediate care for symptoms of lip/oral cavity lump and up to three weeks for unexplained ulceration. ^4^ Based on the recommendation of an appointment within two weeks. ^5^ Based on diagnostic confirmation, test is to be done within 30 days after request and treatment initiation within 60 days after the positive diagnosis.

## Data Availability

The data presented in this study are openly available in Harvard Dataverse at [https://doi.org/10.7910/DVN/8MWMVT], accessed on 29 January 2022.

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
