# Peer review of "Time-to-Treatment of Oral Cancer and Potentially Malignant Oral Disorders: Findings in Malaysian Public Healthcare"

_dentistry, 2022, doi:10.3390/dj10110199_

Round 1

Reviewer 1 Report

This study evaluated the time-to-treatment of oral cancer and oral potentially malignant disorders (OPMD) in a Malaysian public healthcare setting while exploring its contributing factors. It consisted of (1) a cross-sectional patient survey to quantify time to seek care and barriers faced, and (2) a retrospective medical record abstraction to determine treatment and management intervals. The prolonged time-to-treatment of oral cancer and OPMD in Malaysia was still pri- 324 marily contributed by the time taken by patients to seek care.  The findings reiterated the need to reform current initiatives to better promote early lesion recognition by patients and implement strategies for the elimination of their access barriers.

The study is sound and well written. It provides important information on time-to-treat in a Southeast Asian country. I recommend for pulication after revision.

Minor comments:

1. Please provide the sample size calculation and formula.

2. Please explain and/or apply the adjusted-p value methods in statistics because there were multiple comparision in this study: Table 2, 3

3. Please add your study in table 4

Author Response

This study evaluated the time-to-treatment of oral cancer and oral potentially malignant disorders (OPMD) in a Malaysian public healthcare setting while exploring its contributing factors. It consisted of (1) a cross-sectional patient survey to quantify time to seek care and barriers faced, and (2) a retrospective medical record abstraction to determine treatment and management intervals. The prolonged time-to-treatment of oral cancer and OPMD in Malaysia was still primarily contributed by the time taken by patients to seek care.  The findings reiterated the need to reform current initiatives to better promote early lesion recognition by patients and implement strategies for the elimination of their access barriers.

The study is sound and well written. It provides important information on time-to-treat in a Southeast Asian country. I recommend for publication after revision.

  • We appreciate the reviewers' acknowledgment of the importance of this study

  1. Please provide the sample size calculation and formula.
  • We appreciate the feedback on the sample size calculations. We have further added the formula and details applied for the sample size calculations [page 2, line 88-99]

2. Please explain and/or apply the adjusted-p value methods in statistics because there were multiple comparision in this study: Table 2, 3

  • We have added further details on the groups being compared in the footnote of Table 2 and Table 3 [page 4, line 166-167; page 5, line 180-181]

3. Please add your study in table 4

  • We greatly appreciate the suggestion to include our study findings in Table 4. We have added the average intervals accordingly [page 7, line 246]

Reviewer 2 Report

Maybe consider irregular dental visits as one of the reasons for the delay? Dentists are the first ones to notice premalignant lesions during routine dental care. 

Author Response

Maybe consider irregular dental visits as one of the reasons for the delay? Dentists are the first ones to notice premalignant lesions during routine dental care.

  • We agree with the reviewer on the possibility of delays to be contributed by irregular dental visits, as this was also reflected in our study findings. We have added the point raised in our discussion [page 7, line 277-278]

Reviewer 3 Report

Dear Authors,

The article: 'Time-to-Treatment of Oral Cancer and Oral Potentially Malignant Disorders: Findings in Malaysian Public Healthcare' was to quantify time to seek care and barriers faced, and (2) a retrospective medical record abstraction to determine treatment and management intervals.

English language and style must be corrected. Punctuation mistakes should be corrected. 

Abstract: Sentences do not begin with numbers. This is an invalid notation (67.0% of patients attributed delays in seeking care to their inability to recognize potential 30 malignancies).

Introduction is crealy written

Materials and methods:

Add information about the REC in this dection.

Add a table with abbreviations.

References should be prepared in accordance with the MDPI guidelines

To sum up, article should be reconsider after minor revision.

Author Response

The article: 'Time-to-Treatment of Oral Cancer and Oral Potentially Malignant Disorders: Findings in Malaysian Public Healthcare' was to quantify time to seek care and barriers faced, and (2) a retrospective medical record abstraction to determine treatment and management intervals.

English language and style must be corrected. Punctuation mistakes should be corrected.

  • We truly appreciate the feedback. We have proofread and further improved the style and punctuation of our article

Abstract: Sentences do not begin with numbers. This is an invalid notation (67.0% of patients attributed delays in seeking care to their inability to recognize potential malignancies).

  • We have amended the sentence to not begin with numbers and further improved the clarity of the point [page 1, line 30-31]

Introduction is clearly written

  • We appreciate the reviewers' acknowledgment of the Introduction section

Materials and methods:

Add information about the REC in this dection.

  • We believe that this feedback is on the inclusion of the study being approved by MREC (Ministry of Health Medical Research Ethics Committee). To avoid repetition, the ethics approval was mentioned in the ‘Institutional Review Board Statement’ on page 9, line 348-351.

Add a table with abbreviations.

  • We appreciate the feedback. However as the journal format does not include the table of abbreviations and the abbreviations used were minimal for a supplementary attachment, the authors have agreed to have the abbreviation in parentheses after the written-out form only.

References should be prepared in accordance with the MDPI guidelines

  • We appreciate the feedback and acknowledge the irregularities in the referencing format. We have updated the reference according to the Endnote format guide provided by the journal

To sum up, article should be reconsider after minor revision.

  • We would like to thank the reviewer for the constructive comments in further improving our article
